# Applications of artificial intelligence and urban innovation performance: A quasi-natural experiment based on the pilot zones for the innovative application of artificial intelligence

**Shuang Han**◉, **Xianmin Sun**◉◉*, **Shen Zhong**◉

School of Economics, Harbin University of Commerce, Harbin, China

◉ These authors contributed equally to this work.
* sxm@hrbcu.edu.cn

**Data availability statement:** All relevant data are within the paper and its Supporting information files.

## Abstract

Innovation is the core driving force behind the ability to cope with risks and enhance urban competitiveness. Against the backdrop of frequent global crises and intensified competition, improving urban innovation performance is important. This study uses 283 Chinese cities from 2012 to 2023 to empirically analyze the impact of artificial intelligence (AI) applications on urban innovation performance and the underlying mechanism. The results show that AI significantly promotes urban innovation performance while simultaneously improving both the quality and quantity of urban innovation. Moreover, through testing, it was found that the enhancement of human capital and absorptive capacity has become an important channel through which AI drives urban innovation. Additionally, the degree of government intervention weakens the positive impact of AI applications on urban innovation performance. Further research reveals that the impact of AI applications on urban innovation performance varies according to geographical region, implementation time of policies, and scale of cities.

## Introduction

Urban innovation has emerged as a core driving force for enhancing urban competitiveness [1]. Urban innovation performance is used to evaluate the achievements of a city in the field of innovation and serves as a comprehensive evaluation criterion for a city's innovation capabilities [2]. The literature reflects the significance of urban innovation performance. They have a remarkable promoting effect on economic growth [3], environmental protection [4,5], and resource allocation [6]. In regions with high urban innovation performance, there is a high concentration of human capital, which not only strongly propels industrial upgrading [7], but also significantly promotes the continuous growth of total factor productivity [8], thereby laying a solid foundation for local economic development. Moreover, this influence

**Funding:** The author(s) received no specific funding for this work.

**Competing interests:** The authors have declared that no competing interests exist.

was sustainable. In contrast, regions with lower urban innovation performance lack innovation driving forces and rely mostly on local resource advantages for development. This not only imposes a burden on the ecological environment [9] but also makes it difficult to achieve efficient and significant breakthroughs in economic growth. The specific reasons for the difficulty in improving urban innovation performance are as follows: On the one hand, there are challenges in the agglomeration of innovation factors, such as insufficient investment in innovation funds, a shortage of innovative talent, and the irrational allocation of innovation funds, making it difficult to achieve Pareto optimality [10]. However, the urban innovation ecosystem is not perfect, with insufficient coordination among industry, academia, research, and application and low efficiency in the transformation of scientific and technological achievements. Many cutting-edge technologies are difficult to implement on a large scale in the real economy, presenting the "Darwin Sea" effect. Additionally, the economic recession caused by the COVID-19 pandemic has led to the interruption or delay of innovation projects, directly hindering the improvement of urban innovation capabilities and the development of new technologies [11]. Therefore, breaking through the dilemma of improving urban innovation performance and conducting in-depth research on how to enhance it is highly valuable and practical.

Scholars have conducted a series of discussions on the influencing factors and improvement paths of urban innovation performance, focusing on aspects such as urban infrastructure construction [12–14], corporate research and development investment [15], regional collaborative innovation [16,17], the degree of land marketization [18], and government intervention [19]. However, in the post-pandemic era, the effectiveness of the above paths in promoting the growth of urban innovation performance weakened. As Appendix 1 shows, the growth rate of urban innovation performance has declined in recent years. Therefore, it is crucial to seek new ways to improve urban innovation performance in future research. The Solow growth model points out that technological progress is a lasting driving force for long-term economic growth [20].

With the development and application of the new generation of information technology, AI is booming globally, and it also provides new opportunities for urban innovation performance. Existing research shows that AI is expected to become a new engine for promoting growth and improving urban innovation performance [21]. Under the trend of deep integration of AI and traditional industries, AI can promote the agglomeration of production factors and enhance the level of human capital and technological innovation capabilities [22]. This type of agglomeration can generate technological and knowledge spillovers, enabling enterprises to strengthen the external technological spillover effect, optimize the reallocation of resources, and improve the efficiency of internal resource reallocation [23], thereby promoting the improvement of urban innovation performance. Existing literature has discussed the impact of AI on urban innovation performance through the application of industrial robots [24] and enterprise production [25]. However, the selected perspectives are relatively singular, and they cannot measure the comprehensive impact and net effect of AI on urban innovation performance. In 2019, China officially launched a pilot zone policy for the innovative application of AI, which provided an excellent quasi-natural experiment for studying the impact of AI on urban innovation performance.

Therefore, this study establishes a leading area for the innovative application of AI as an exogenous shock variable. Using data from 283 cities in China from 2012 to 2023 as a sample and adopting the multi-period difference-in-differences (DID) model, we explore the impact of AI applications on urban innovation performance. The remainder of this paper is structured as follows. Section 2 reviews the relevant literature. Section 3 presents the policy background and theoretical analysis and proposes the research hypotheses. Section 4 introduces

the research methods and design. Section 5 analyzes urban innovation performance in China. Section 6 conducts an empirical test on the data and explores the impact mechanism and the heterogeneity of the impact. Section 7 presents the conclusions and policy recommendations of the study.

## Literature review

Schumpeter's seminal 1912 work, The Theory of Economic Development, contains pertinent discussions on innovation. He noted that innovation is the establishment of a new production function and a new combination of production factors. On this basis, Cooke proposed that in a specific institutional environment, a regional organization composed of geographically proximate innovation entities with a mutual learning relationship forms a regional innovation system [26]. In recent years, scholars have focused mostly on the factors influencing urban innovation performance. Open innovation actions [27,28], the synergy of innovation factors [29,30], knowledge management [31,32], the intelligence of the manufacturing industry [33] and digital transformation [34] can all affect urban innovation performance.

In addition, the role of government behavior in influencing urban economic development has been widely confirmed [35]. Policy guidance, financial support [36,37], and R&D subsidies [38,39] are the main manifestations of the government's role in scientific and technological innovation. However, on the basis of these studies, determining the net effect of government interventions on urban innovation performance is difficult. Therefore, an increasing number of scholars have selected specific policies formulated by the government to construct quasi-natural experiments, adopted the DID model, and focused on the causal relationship between innovation policies and innovation performance. Smart city construction policy is a widely discussed topic among scholars. The construction of smart cities is beneficial to urban innovation performance [40], and its promotion effect is sustainable. With improvements in the level of economic development and information infrastructure, urban innovation performance will also increase [41]. Moreover, this promotional effect has a technological spillover effect [42]. Furthermore, government intervention, represented by the national innovative city pilot policy, can significantly improve the urban innovation level [43], optimize the allocation of innovation factors [44], and have a long-term impact on the innovation performance of local and surrounding areas. This spillover effect is magnified as innovation performance improves [19]. This policy will also improve energy utilization efficiency [45], promote green technology innovation [46], help reduce carbon emissions, and promote low-carbon and sustainable development [47,48]. In the post-pandemic era, the growth of science and technology in innovation and the economy has attracted increasing attention from scholars and the government.

As the main engine of the Fourth Industrial Revolution, the concept of AI was officially introduced at the Dartmouth Conference in 1956, marking the initiation of research in this field [49]. Numerous studies have shown that there are numerous opportunities to fully unleash the innovative potential of AI [50]. AI provides opportunities for transforming the data generated in digital systems into new applications and enhancing operational efficiency [51]. Moreover, AI can promote the agglomeration and upgrading of human capital [52,53], generate knowledge spillover effects [54], and provide a foundation for innovation. Moreover, AI can improve regional innovation performance by optimizing the allocation of innovation factors, and AI can improve regional innovation performance [55]. The application of intelligent machinery such as robots has increased labor productivity and reshaped the nature of the research and development innovation process and organizations. From a technology-oriented

perspective, the application of intelligent machinery is an unconventional source of innovation [56], thus promoting the improvement of urban innovation performance. Specifically, AI is also a driving force for organizational innovation performance [57] and green technology innovation [58].

Although scholars have conducted extensive discussions on the benefits at the production end of AI, intelligent applications, and the resulting environmental and management benefits, existing research still has the following deficiencies. First, research analyzing the impact of AI-related policies on urban innovation performance from the perspective of government policy intervention is lacking. Second, the changes in urban innovation performance before and after government intervention have not been considered; that is, it is hard to determine the net effect of government intervention on urban innovation performance. Third, there is a lack of discussion on the impact of AI-related policies on urban innovation performance.

Therefore, this study makes the following contributions. First, the policy of the pilot zone for the innovative application of AI was selected to explore its impact on urban innovation performance, which enriches the influencing factors of urban innovation performance. Additionally, this study combined the application of AI with government behavior to explore its economic benefits. Second, this study utilized innovation performance data at the city level in China. Taking into account multiple batches of pilot cities, a multi-period DID model is employed to evaluate the policy effects of the pilot zone for the innovative application of AI and to examine the net benefits of AI on urban innovation performance. Furthermore, this study analyzes the mechanism of the policy of the pilot zone for the innovative application of AI to promote urban innovation performance, and studies in detail how AI application affects urban innovation performance.

## Development background and theoretical hypotheses

### Development background

Since the beginning of the 21st century, the development of the digital economy and emerging technologies has accelerated continuously. Globally, the governments of various countries have successively incorporated AI development strategies into their top-level designs and actively promoted the implementation and application of related technologies.

The United States had an early layout in the field of AI. In 2019, it launched the "American AI Initiative". In the same year, the European Commission released its White Paper on AI and officially launched a tender for the establishment of AI factories. The British government also announced the "AI Sector Deal: Action Plan for AI", aiming to create areas of AI growth. Moreover, AI application policies in Asia have received increasing attention. In 2019, Japan introduced the "AI Strategy 2019", and the Ministry of Science and ICT of South Korea released the "National AI Strategy" and the Digital New Deal Promotion Plan.

Similarly, starting in 2019, the Chinese government established three batches of AI innovations and application pilot zones. As of the end of 2023, a total of 11 cities in China have been involved in the planning and construction of these zones, aiming to achieve remarkable results in aspects such as breaking through key technologies, constructing an innovative ecosystem, and realizing the integration of "AI +". The establishment time and development strategies of each AI innovation and application pilot zone in China are presented in Appendix 1. Compared with non-pilot cities, pilot cities have achieved relatively rapid development in dimensions such as the gathering of high-end talent and the construction of intelligent infrastructure, injecting impetus into the intelligent upgrading of regional industries, the breakthrough of scientific and technological innovation, and making contributions to the improvement of urban innovation performance and economic development. The specific

implementation of AI innovation application policies in each leading zone can be found in the Appendix.

## Theoretical hypotheses

**Direct effects.** First, according to Schumpeter's innovation theory, AI, as a new disruptive technology, accelerates the process of "creative destruction" [59]. Enterprises can rapidly iterate and develop new technologies through AI, shorten the innovation cycle, promote a multiplier effect on urban innovation output, and increase urban innovation.

Second, the application of AI has promoted the transformation of traditional production factors. In terms of labor factors, AI not only changes the labor supply structure through automation substitution [60], but also improves the efficiency of skill learning through auxiliary training systems such as machine learning [61], pushing the labor supply curve to shift continuously to the right and forming a new high-skill-oriented equilibrium. An increase in high-skilled labor can directly promote urban innovation output, reduce the costs of knowledge diffusion and learning, and provide a foundation for innovation [62]. In addition, with respect to capital factors, the application of AI, such as industrial robots, promotes an increase in capital–labor substitution elasticity and expands the production possibility frontier. Moreover, when AI technologies such as algorithms are used, asset allocation can be optimized, and the marginal efficiency of capital can be improved [63]. This improvement breaks the law of diminishing the marginal returns of traditional capital accumulation, stimulates enterprises to expand the scale of innovation investment, and makes high-risk and breakthrough innovations feasible, thus promoting the improvement of urban innovation quality.

Furthermore, the interaction between these two mechanisms gives rise to the upgrading of the innovation ecosystem. The improvement in capital efficiency attracts the agglomeration of high-skilled talent, and the agglomeration of talent further enhances the efficiency of capital use. This virtuous cycle forms a favorable innovation ecosystem, which ultimately results in a qualitative change in urban innovation ability. On this basis, Hypothesis 1 is proposed.

Hypothesis 1: AI application directly enhances urban innovation performance in terms of both "quality" and "quantity".

**Upgrading of human capital.** The application of AI can promote the upgrading of human capital, which boosts urban innovation performance. First, we consider the impact of AI applications on the upgrading of human capital. Compared with traditional machinery, AI has a greater degree of automation and a greater substitution effect on labor [64]. Moreover, with the application of AI, the phenomenon of "employment polarization" emerges, increasing the demand for high-skilled labor that is compatible with technology [65]. At the same time, AI affects existing human labor fields in four ways: substitution, enhancement, adjustment, and reconstruction. These impacts lead to employment migration in the labor force and require workers to improve their human capital levels to meet higher demands [66]. Additionally, the deep integration of AI technology and the production process can drive the transformation of the labor force structure from physical labor to mental labor and from low-skilled to high-skilled, directly and efficiently promoting the improvement of human capital levels and giving full play to individual values, thus facilitating the achievement of innovation outcomes [60].

Human capital is a source of innovation [67]. The participation of human capital in innovation activities is the core element of realizing innovation [68,69]. First, compared with low-skilled labor, high-skilled labor has stronger learning and creative abilities, providing more possibilities and guarantees for enterprises within the city to adopt new equipment and conduct research and development of new technologies [70]. Second, human capital enhances

innovation performance through the knowledge spillover effect. Human capital has several externalities. High-quality human capital can promote innovation exchanges and cooperation among regions, facilitate the spillover and diffusion of knowledge, and drive technological progress, thereby improving urban innovation performance [71,72]. On this basis, we propose Hypothesis 2.

Hypothesis 2: The application of AI promotes human capital upgrades, thereby improving urban innovation performance.

**Enhancement of absorptive capacity.**   The application of AI can enhance a city's absorptive capacity, which, in turn, boosts urban innovation performance. Absorptive capacity refers to the ability to identify, acquire, and utilize knowledge from the external environment [73]. The application of AI can accelerate the transformation and utilization of new knowledge, break the old cognitive structure, and enable understanding and adaptation to new knowledge that is incompatible with original knowledge [74]. Moreover, the application of AI has transformed traditional information processing methods, shortened the time for knowledge acquisition, expanded the acquisition channels, made the reception of new knowledge more efficient, and thus improved absorptive capacity.

Absorptive capacity largely determines enterprises' innovation ability and vitality. Strong absorptive capacity enables enterprises to transform the knowledge learned beyond organizational boundaries into resources that can be utilized by themselves [75]. This expands their knowledge base and provides opportunities for innovation in new technologies. Moreover, regions with strong absorptive capacity have a greater ability to restructure and expand knowledge, improve the efficiency of knowledge utilization, and further increase the quantity of urban innovation. On this basis, Hypothesis 3 is proposed.

Hypothesis 3: The application of AI promotes urban innovation performance by enhancing absorptive capacity.

**Moderating effect of the degree of government intervention.**   Government intervention is crucial for improving urban innovation performance [19]. Neither excessive intervention nor overly lenient policies improve urban innovation performance. First, from the perspective of government supervision, on the one hand, government supervision has a policy constraint effect on the entire capital market, thereby increasing the supervision intensity of venture capital institutions over innovative projects. While reducing venture capital, it also inhibits the technological innovation of enterprises [76], thus weakening the promoting effect of AI applications on urban innovation performance. However, during the process of AI applications promoting urban innovation performance, excessive supervision increases innovation costs in various ways [77]. Stringent supervision requires enterprises to invest large amounts of resources to meet compliance standards during the project development process [78], which increases the costs of technological research, development, and operations and weakens the vitality of innovation.

Second, from the perspective of financial support, when the government simultaneously implements an expansionary fiscal policy and strongly advocates scientific and technological innovation, it leads to an increase in public R&D expenditures. Competition for innovation resources and rent-seeking behaviors among innovation entities will intensify, and social research and development costs will rise, thus reducing urban innovation ability [79]. However, excessive financial subsidies and capital support make innovation entities develop a sense of dependence, suppress their enthusiasm for innovation, and weaken the promoting effect of AI applications on urban innovation performance [80].

Furthermore, from the perspective of market access, excessive government intervention raises the threshold for market access, making it difficult for many small start-up companies and innovative teams to enter. This reduces the number of market participants and the degree

of market competition, which is not conducive to giving full play to the promoting effect of AI on urban innovation performance and stimulating the vitality of innovation [81]. On this basis, we propose Hypothesis 4.

Hypothesis 4: The degree of government intervention weakens the positive effect of AI applications on urban innovation performance.

On the basis of the above analysis, the mechanism diagram of this study is shown in Fig 1.

## Research design

### Model construction

**Multi-period DID model.** In the process of promoting the establishment of pilot zones for the innovative application of AI, various regions have focused mainly on the deep integration of AI and the real economy. The establishment of pilot zones for the innovative application of AI will lead not only to differences in innovation performance between pilot cities and non-pilot cities before and after the implementation of the policy but also to systematic differences in the individual characteristics of pilot cities and non-pilot cities, resulting in differences in urban innovation performance between pilot and nonpilot cities at the same point in time. Therefore, by establishing pilot zones for the innovative application of AI as a quasi-natural experiment, a multi-time-point DID model is constructed to evaluate the impact of AI applications on urban innovation performance. Accordingly, this study regards the pilot cities as the treatment group and the non-pilot cities as the control group and constructs the following multi-time-point DID model:

$$UIP_{it} \begin{cases} Iquality_{it} \\ Iquantity_{it} \end{cases} = \alpha_0 + \alpha_1 AI\_Policy_{it} + \alpha_2 \sum Controls_{it} + \gamma_i + \delta_t + \varepsilon_{it} \quad (1)$$

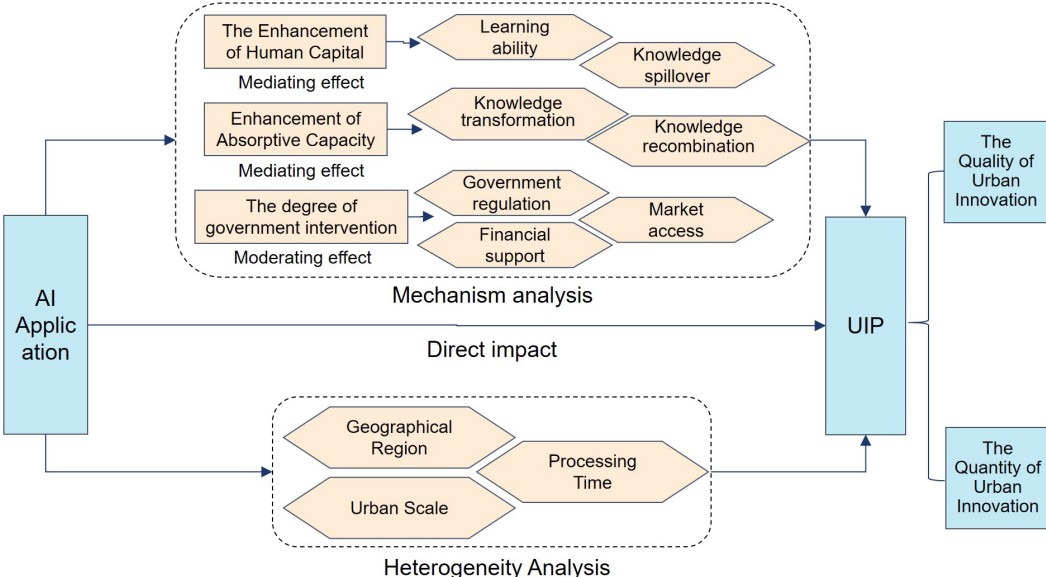

**Fig 1. Mechanism diagram.**

where i and t represent the city and year, respectively. The explained variable $UIP_{it}$ is the performance of urban innovation, which is manifested as the innovation index of city i in year t, and $Iquality_{it}$ and $Iquantity_{it}$ represent the quality of urban innovation and the quantity of urban innovation, respectively. $AI\_Policy_{it}$ is the establishment of the pilot zone for the innovative application of AI, which is the core explanatory variable of this study; $\sum Controls_{it}$ is the set of control variables; $\gamma_i$ is the urban fixed effect; $\delta_t$ is the time fixed effect; and $\varepsilon_{it}$ is the random disturbance term. The estimated coefficient measures the average difference in urban innovation performance before and after the establishment of the pilot zone for the innovative application of AI.

**Mediating effect model.** According to previous research, the level of human capital (Lhuca) and absorptive capacity (Cabsorp) are important channels through which AI promotes urban innovation. A mediating effect model was constructed to test this impact pathway.

$$X_{it} = \beta_0 + \beta_1 AI\_Policy_{it} + \beta_2 \sum Controls_{it} + \gamma_i + \delta_t + \varepsilon_{it} \qquad (2)$$

$$UIP_{it} \begin{cases} Iquality_{it} \\ Iquantity_{it} \end{cases} = \gamma_0 + \gamma_1 AI\_Policy_{it} + \gamma_2 X_{it} + \gamma_3 \sum Controls_{it} + \gamma_i + \delta_t + \varepsilon_{it} \qquad (3)$$

Among these variables, $X_{it}$ serves as the mediating variable, which in this study represents the level of human capital and absorption capacity. $\beta_1$ measures the impact of AI applications on the mediating variable, and $\gamma_1$ represents the impact of AI applications on urban innovation performance after taking the mediating variable into account.

**Moderating effect model.** The empirical results presented earlier confirm that the degree of government intervention weakens the promoting effect of AI applications on urban innovation performance. To test this hypothesis, the following model was constructed:

$$UIP_{it} \begin{cases} Iquality_{it} \\ Iquantity_{it} \end{cases} = \rho_0 + \rho_1 AI\_Policy_{it} + \rho_2 AI\_Policy_{it} \times Dinter_{it} + \rho_3 \sum Controls_{it} \qquad (4)$$

$$+ \gamma_i + \delta_t + \varepsilon_{it} \qquad (5)$$

Among them, $AI\_Policy_{it} \times Dinter_{it}$ served as the moderating factor, and $\rho_2$ moderated the intervention effect.

## Variable selection and data explanation

**Explained variable: Urban Innovation Performance (UIP).** Currently, there are four main types of indicators for measuring urban innovation performance. First, the number of patent applications is used for measurement. Although the number of patent applications can better reflect the true level of urban innovation than can the number of granted patents, the use of the number of patent applications of micro-enterprises to measure macro-level urban innovation ability is relatively one-sided and is affected by bureaucratic factors. Second, an indicator system is used to measure urban innovation performance. Although it is more comprehensive than the number of patent applications, the selection of indicators and evaluation methods lacks a unified standard, which is likely to cause deviations in the measurement results of variables. Third, the input–output efficiency value of urban innovation is used to calculate urban innovation performance. This method is also relatively comprehensive, but it has

difficulty covering all aspects of indicator selection. Fourth, the urban innovation index from the Report on China's Urban and Industrial Innovation Capabilities is employed to measure urban innovation capabilities. This index calculates patent values via a patent model, thereby circumventing potential issues with the aforementioned indicators and enabling a relatively objective reflection of the actual innovation capabilities of each city. Therefore, this study uses the urban innovation index in the "Report on China's Urban and Industrial Innovation Capabilities" to measure urban innovation capabilities. To facilitate comparison and research, we conducted standardization processing and then incorporated it into the regression model. Referring to Li X (2023) [82], the number of invention patents granted per capita is selected to measure the quality of urban innovation (Iquality), and referring to Zhang M (2020) [83], the number of patents granted per capita in the region is selected to measure the quantity of urban innovation (Iquantity).

**Explanatory variable: Applications of AI (AI_Policy).** This study takes the policy of the pilot zone for the innovative application of AI as a quasi-natural experiment and uses the interaction term of the urban-type dummy variable and the policy implementation time dummy variable to represent the policy treatment effect of the policy of the pilot zone for the innovative application of AI to represent the situation of AI application. Specifically, this study set the number of pilot cities in the pilot zone for the innovative application of AI to 1, which served as the experimental group, and set the number of non-pilot cities to 0, which served as the control group. The time dummy variables before and after the implementation of the pilot policy were set to 0 and 1, respectively. In the existing literature, pilots that start in the first half of the year in the multi-period DID are regarded as starting in the current year, and pilots that start in the second half of the year are regarded as starting in the next year. Therefore, for the samples of Shanghai from 2019 and later; Jinan, Qingdao, and Shenzhen from 2020 and later; and Beijing, Tianjin, Hangzhou, Guangzhou, and Chengdu from 2021 and later, the value is assigned as 1; for the remaining samples, the value is 0.

**Control variables.** On the basis of literature, the control variables affecting urban innovation performance include the regional economic development level (GNP), measured by per capita GDP; Population density (Pdensity), the ratio of the population in the region to the land area; Degree of openness to the outside world (Dopen), measured by the ratio of the amount of foreign investment to the regional gross domestic product; Infrastructure level (Vpostal), measured by the total volume of postal services; Informatization level (Vtele), measured by the total volume of telecommunications services; and Scientific and technological development level (Npetent), measured by the number of patent applications. Fiscal decentralization (DFiscal): The proportion of local fiscal expenditure in total fiscal expenditure is selected.

**Mediating variables.** The above theoretical analysis reveals that the levels of human capital (Lhuca) and absorption capacity (Cabsorp) are important channels through which AI applications affect urban innovation performance. Therefore, this study selects the number of R&D personnel per 10,000 people to measure the level of human capital related to innovation, selects expenditures on science and technology to measure absorption capacity, and then explores its mediating role.

**Moderating variable.** On the basis of the above analysis, the degree of government intervention (Dinter) plays a reverse moderating role in the impact of AI application on urban innovation performance. Therefore, this study selected the ratio of general government fiscal expenditure to regional gross domestic product to measure the degree of government intervention.

## Sample sources and descriptive statistical analysis

This study selected 283 prefecture-level cities in China between 2012 and 2023 as the research objects. The data are sourced from the "Report on China's Urban and Industrial Innovation Capabilities", the "China City Statistical Yearbook", the local statistical yearbooks of each city, and the statistical bulletins of various governments. Descriptive statistical analyses of the main variables are presented in Table 1.

## Analysis of the current situation of innovation performance in various cities

Fluctuations in urban innovation performance are influenced by several factors. Because Nanjing, Changsha, and Wuhan were launched relatively late and were not within the coverage of the sample, the pilot cities in 2019, 2020, and 2021 were selected as the samples for analysis. The analysis of the innovation performance of each pilot city in the leading zones for artificial intelligence innovation applications is as follows:

### Analysis of the growth of innovation performance in pilot cities

On the basis of the calculated growth rates of urban innovation performance in Table 2 and Fig 2, the following analysis is conducted. First, the growth rates of all the pilot cities are greater than zero, and the innovation index shows an upward trend. Simultaneously, the growth rate of the innovation index fluctuated, and there were significant differences among the cities. In some years, the growth rates of certain cities became relatively prominent. For example, in 2015, the growth rate of Qingdao's innovation index reached 0.485, and the growth rates in 2015 and 2016 exceeded 40% for two consecutive years, indicating rapid growth.

The pilot cities were analyzed according to differences in the implementation times of the policies. After the policy was implemented in 2019, Shanghai's innovation index growth rate decreased slightly in 2020 but then increased in 2021. It is likely that, in the initial stage of policy implementation, industrial transformation and innovation activities require time accumulation, and policy dividends are not fully released. For Shenzhen, Jinan, and Qingdao, where the pilot policies were implemented in 2020, the growth rates of the innovation index increased. The growth rate of Shenzhen's innovation index increased from 0.183 to 0.227, Jinan's from 0.190 to 0.244, and Qingdao from 0.223 to 0.239, which, to a certain extent, shows that the policies stimulated innovation vitality to some extent. After the implementation of pilot city policies in 2021, the innovation indices of some cities changed to a certain extent. For example, Guangzhou's innovation index rose to 0.305 in 2021 but then fluctuated

**Table 1. Descriptive statistical analysis of the main variables.**

| Variables | Obs | Mean | S.D. | Min | Max |
|---|---|---|---|---|---|
| UIP | 3396 | 37.910 | 166.120 | 0.020 | 3956.430 |
| AI_Policy | 3396 | 0.010 | 0.100 | 0.000 | 1.000 |
| GNP | 3103 | 5.820 | 3.420 | 0.820 | 25.690 |
| Pdensity | 3098 | 5.730 | 0.990 | 1.740 | 9.090 |
| Dopen | 3045 | 0.190 | 0.580 | 0.000 | 28.370 |
| Vpostal | 3113 | 21.250 | 72.510 | 0.140 | 1966.940 |
| Vtele | 3113 | 48.530 | 81.740 | 0.000 | 1396.400 |
| Npetent | 3103 | 1.090 | 2.820 | 0.000 | 36.020 |
| DFiscal | 3113 | 0.450 | 0.220 | 0.060 | 1.540 |

**Table 2. The growth rate of innovation performance of the pilot cities.**

| Period | 2019 | 2020 | | | 2021 | | | | |
|---|---|---|---|---|---|---|---|---|---|
| Year | Shanghai | Shenzhen | Jinan | Qingdao | Beijing | Tianjin | Hangzhou | Guangzhou | Chengdu |
| 2013 | 0.244 | 0.254 | 0.286 | 0.395 | 0.265 | 0.214 | 0.217 | 0.286 | 0.257 |
| 2014 | 0.199 | 0.203 | 0.270 | 0.379 | 0.231 | 0.211 | 0.198 | 0.240 | 0.236 |
| 2015 | 0.242 | 0.222 | 0.294 | 0.485 | 0.275 | 0.249 | 0.255 | 0.251 | 0.295 |
| 2016 | 0.251 | 0.187 | 0.293 | 0.450 | 0.274 | 0.240 | 0.233 | 0.264 | 0.288 |
| 2017 | 0.219 | 0.182 | 0.259 | 0.280 | 0.256 | 0.235 | 0.243 | 0.296 | 0.249 |
| 2018 | 0.188 | 0.150 | 0.181 | 0.238 | 0.207 | 0.175 | 0.212 | 0.258 | 0.193 |
| 2019 | 0.177 | 0.202 | 0.180 | 0.256 | 0.217 | 0.157 | 0.212 | 0.251 | 0.200 |
| 2020 | 0.173 | 0.183 | 0.190 | 0.223 | 0.214 | 0.156 | 0.261 | 0.251 | 0.206 |
| 2021 | 0.206 | 0.227 | 0.244 | 0.239 | 0.220 | 0.164 | 0.268 | 0.305 | 0.259 |
| 2022 | 0.171 | 0.185 | 0.196 | 0.193 | 0.180 | 0.141 | 0.211 | 0.233 | 0.206 |
| 2023 | 0.146 | 0.156 | 0.164 | 0.162 | 0.153 | 0.124 | 0.174 | 0.189 | 0.171 |

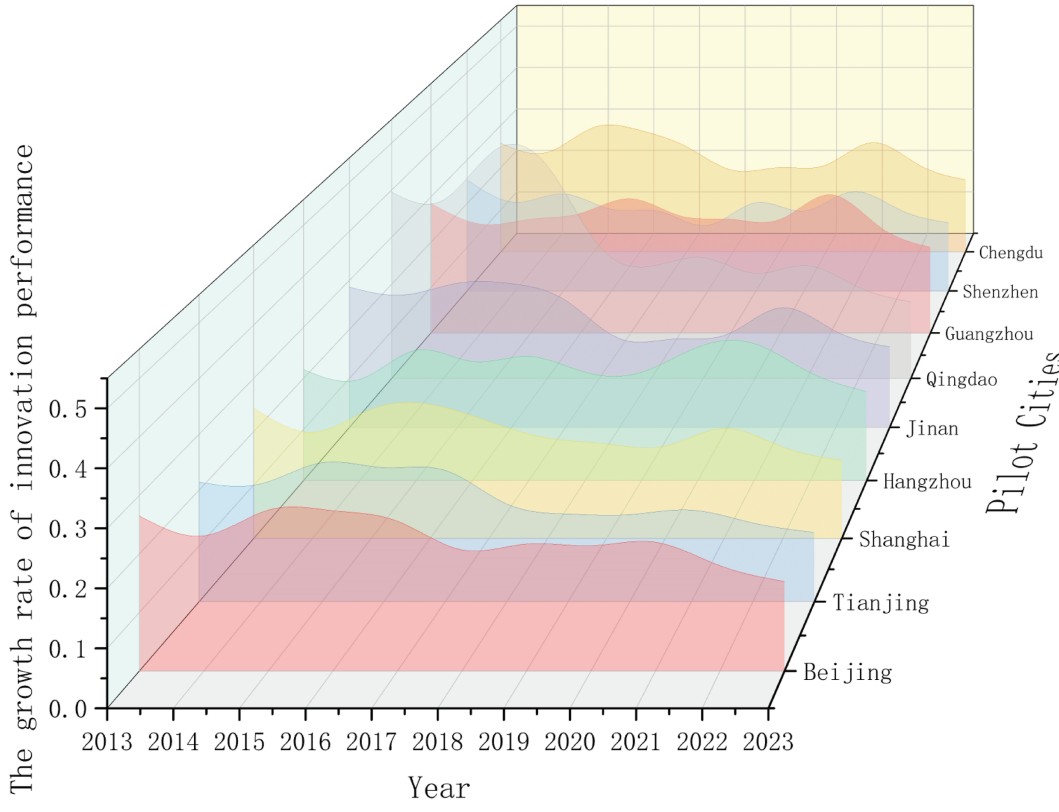

**Fig 2. Fluctuations in the UIP growth rates of the pilot cities.**

and decreased. However, overall, the effect of policies on enhancing the innovation indices of these cities is not yet sufficiently stable. This may be due to the constraints of factors such as the original industrial structure and resource allocation during the implementation of the policies. Moreover, after 2021, the growth rates of the innovation indices of all the pilot cities showed a slowing trend. Behind this phenomenon, the impact of the COVID-19 pandemic

is most likely a key factor. Under the pandemic, global market demand has shrunk, enterprises have cut their R&D investments, and with the disruption of the supply chain, the production and delivery of innovative products have become difficult, which in turn has led to a slowdown in the growth rate of the innovation index.

## Comparison of innovation performance in different regions

The entire country was divided into three regions for analysis. According to the information reflected in Fig 3, from the perspective of the whole country and the three major regions from 2012 to 2023, the average value of innovation performance showed a continuous upward trend over the 12-year period. Growth was relatively slow from 2012 to 2023. Since 2016, the growth rate has significantly accelerated, indicating that over time, the increasing emphasis on and investment in innovation nationwide has gradually yielded more remarkable results. The innovation environment has been continuously optimized, driving the continuous improvement of the innovation index. By 2023, the average values of the entire country and eastern region have reached relatively high levels. The average annual value in the eastern region was significantly greater than that in the central and western regions, and the growth rate was relatively high. After 2016, the growth slope increased, and innovation development accelerated. The values in the central and western regions were relatively close, with those in the central region being slightly greater than those in the western region; however, the difference between the two was not significant, and there was still room for improvement.

Simultaneously, the policy of the pilot zones for innovative application of AI implemented in 2019 had a significant effect on the change in this trend. After the implementation of the policy in 2019, the growth rate of the national urban innovation index accelerated significantly. It promoted the deep integration of AI and various industries nationwide; guided

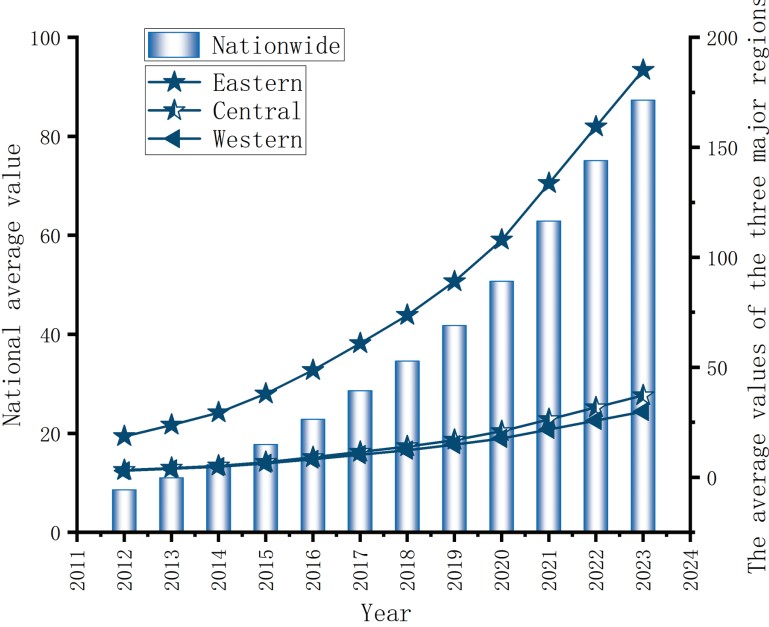

**Fig 3. The changing trend of innovation performance.**

localities to increase investment in the research, development, and application of AI; optimized the allocation of innovation resources; and facilitated the improvement of innovation capabilities in various regions through the use of AI technologies.

## Empirical analysis

### Preliminary testing

**Policy effects of the pilot zones for the innovative application of AI.**  To ensure the logical validity of this study, the precondition must first be met; that is, the policy of the pilot zones for the innovative application of AI has effectively improved the level of AI application in pilot cities. Therefore, this study examined the differences in the level of AI application in each city before and after the implementation of the pilot zone policy for the innovative application of AI. In accordance with Shen et al. (2024) [84], the stock of AI enterprises (AI-Enterprise) and the density of robot stock (AI-Robot) are used to measure the degree of AI application in each city, and these are used as the explained variables for regression analysis. The results are presented in Table 3. The regression results show that the estimated coefficients of AI_Policy are all significantly positive, indicating that, after the implementation of the policy, there is a significant improvement in the level of AI application of enterprises in the pilot areas. This finding also shows that using the policy of pilot zones for the innovative application of AI as an exogenous shock variable has a high degree of credibility.

**Parallel trend test and analysis of dynamic effects.**  The prerequisite for using the multi-period DID model is that the experimental and control groups maintain a consistent trend of change before policy implementation. Because pilot cities receive policy shocks at different times, it is not possible to simply take a certain year as the critical point of policy implementation to set the event variable. Instead, it was necessary to set a dummy variable representing the relative time value of the implementation of innovative city policies for each pilot city. The formula for the parallel trend test is as follows:

$$UIP_{it} = \phi_0 + \phi_1 \sum_{-9}^{3} D_t + \phi_2 \sum Controls_{it} + \gamma_i + \delta_t + \varepsilon_{it} \tag{6}$$

Within this context, $D_t$ represents the annual dummy variable distinguishing the periods before and after policy implementation, whereas $\phi_1$ serves as the coefficient of interest. This coefficient assesses whether the treatment and control groups exhibited parallel trends prior to the establishment of the pilot zone for the innovative application of AI. The remaining variables were incorporated into the model, which is consistent with the multi-time point DID model framework. As shown in Fig 4, the results reveal that the coefficients of the dummy

**Table 3. Results of the preparatory inspection.**

| Variable | AI-Enterprise | AI-Robot |
|---|---|---|
| Columns | (1) | (2) |
| AI_Policy | 2.030*** | 7.612** |
|  | (0.055) | (3.067) |
| City FE | YES | YES |
| Year FE | YES | YES |
| Observations | 3,026 | 3,026 |
| R-squared | 0.671 | 0.814 |

Note: Standard errors are in parentheses. $^{***}p < 0.01$, $^{**}p < 0.05$, and $^{*}p < 0.1$. The same are as blow.

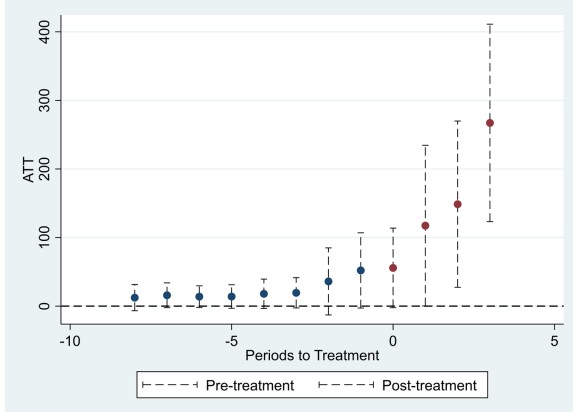

**Fig 4. Parallel trends diagram.**

variables for the relative time before policy implementation are not significant. In the year the policy was implemented, the effect of the pilot policy has not yet been stable. However, one year after implementation, the impact coefficient of the pilot zone policy for innovative AI applications was significantly positive and continued to increase. This finding indicates that the pilot zone policy for the innovative application of AI can generate a policy effect that promotes urban innovation, but there may be a potential lag.

## Baseline regression

Table 4 reports the baseline regression results of the impact of AI applications on the innovation performance of cities. Specifically, the first column presents the estimation results without the inclusion of control variables, and Column (2) shows the estimation results after incorporating the control variables. The results indicate that, regardless of the situation, the coefficient of AI_Policy is significantly positive and passes the significance test at the 1% level. To a certain extent, this demonstrates that AI applications significantly enhance the innovation performance of cities. This suggests that the implementation of AI applications provides a favorable policy environment for the pilot cities, promotes the upgrading of human capital and financial agglomeration, offers a sound institutional environment for innovation and entrepreneurship, and contributes to the improvement of urban innovation performance. Furthermore, an in-depth analysis of the enhancement of both the "quality" and

**Table 4. The results of the baseline regression.**

| Variable | UIP | | Iquality | Iquantity |
|---|---|---|---|---|
| | (1) | (2) | (3) | (4) |
| AI_Policy | 3.789*** | 2.189*** | 1.663*** | 1.189*** |
| | (0.093) | (0.084) | (0.114) | (0.111) |
| Control | NO | YES | YES | YES |
| City FE | YES | YES | YES | YES |
| Year FE | YES | YES | YES | YES |
| Constant | −0.177*** | 0.008 | 0.050* | −0.231*** |
| | (0.027) | (0.022) | (0.030) | (0.029) |
| Observations | 3,396 | 3,026 | 3,026 | 3,026 |
| R-squared | 0.398 | 0.598 | 0.306 | 0.709 |

"quantity" of urban innovation performance by AI reveals that, as shown in Columns (3) and (4) of Table 4, the estimated coefficients of both the quality and quantity of urban innovation are significantly positive at the 1% level. This finding indicates that AI applications are conducive to improving both the "quality" and "quantity" of urban innovation. Hypothesis 1 is thus verified.

### Robustness test

**Changing the estimation method.** To address the issue that samples receiving treatment earlier become the control group for samples receiving treatment later and to avoid the estimated coefficients of the two-way fixed effects model from being affected by cross-period cross-contamination, this study adopts the methods proposed by Goodman-Bacon (2021), Dube et al. (2023), Callaway and Sant Anna (2021), and Cengiz (2019) for robustness under heterogeneous treatment [85–88]. The results are shown in Fig 5 and Tables 5, 6, 7, and 8. According to the results of the Goodman–Bacon test, among all the samples, the proportion

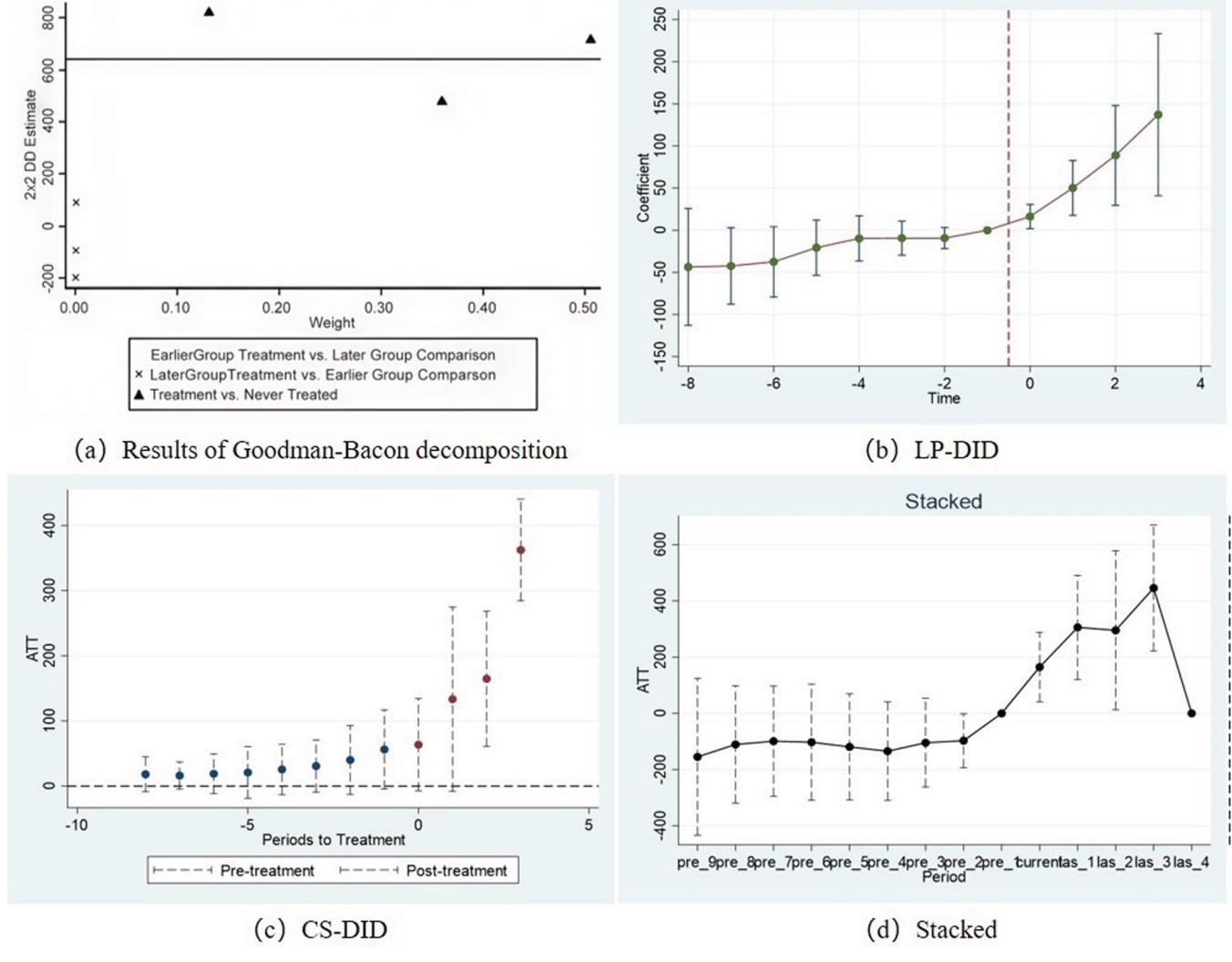

**Fig 5. Test results of changing the estimation method.**

**Table 5. The weighting situations of the AI_Policy coefficient.**

| DD Comparison | Weight | Avg DD Est |
|---|---|---|
| Earlier T vs. Later | 0.003 | −9.188 |
| Later T vs. Earlier | 0.001 | −11.357 |
| T vs. Never treated | 0.996 | 643.892 |

**Table 6. Results before and after the policy under the LP method.**

|  | Coeff. | SE | t | P>t | 95% CI |  | Obs |
|---|---|---|---|---|---|---|---|
| Pre | −29.097 | 23.393 | −1.240 | 0.215 | −75.150 | 16.956 | 802 |
| Post | 82.302 | 28.970 | 2.840 | 0.005 | 25.274 | 139.331 | 2184 |

**Table 7. The results of the CS-DID.**

| Variable | Inno | | | | |
|---|---|---|---|---|---|
| Models | CS-DID | | | | |
|  | Simple-ATT | GAverage | CAverage | Pre-avg | Post-avg |
| Columns | (1) | (2) | (3) | (4) | (5) |
| AI_Policy | 0.750*** | 0.729** | 0.624*** | 0.172 | 1.132*** |
|  | (0.289) | (0.331) | (0.203) | (0.116) | (0.212) |
| Control | YES | YES | YES | YES | YES |
| City FE | YES | YES | YES | YES | YES |
| Year FE | YES | YES | YES | YES | YES |
| Observations | 3,396 | 3,396 | 3,396 | 3,396 | 3,396 |

**Table 8. Analysis of dynamic effects.**

| Stacked DID | | | |
|---|---|---|---|
| Periods | P-value | Periods | P-value |
| D-9 | 0.305 | D-3 | 0.196 |
| D-8 | 0.336 | D-2 | 0.144 |
| D-7 | 0.347 | D0 | 0.009 |
| D-6 | 0.345 | D1 | 0.001 |
| D-5 | 0.226 | D2 | 0.035 |
| D-4 | 0.134 | D3 | 0.000 |

of negative weights is less than 1%, which means that the estimated results are not affected by negative weights. Moreover, the proportion of the results of the treatment group and the group that had never received treatment reached 99.6%, indicating that there would be no time-varying heterogeneous treatment effects in this group. Furthermore, the results of Dube et al. (2023) show that before the implementation of the policy of the pilot zone for innovative application of AI, the estimated coefficient is not significant, with a P value of 0.2146. After the implementation of the policy, the estimated coefficient was 0.0048; the parallel trend plot is shown in Fig 5b. All the results passed the parallel trend test, indicating that the original results were robust. In addition, the parallel trend results obtained by the other methods are approximately the same as the trend of the baseline regression. Before the experiment, there was no significant difference between the experimental group and the control group, satisfying the assumption of parallel trends. After the establishment of the pilot zone, urban innovation performance was significantly greater than that of nonpilot cities, enhancing the robustness of this study.

**PSM-DID.** Although the pilot zone for innovative application of AI, as an exogenous shock event, has largely alleviated the endogeneity problem, the pilot cities of the pilot zone for innovative application of AI may not be completely randomly selected, which, to some extent, increases the inaccuracy of policy evaluation. Simultaneously, considering the heterogeneity of the cities themselves, differences in urban characteristics between the experimental and control groups may also lead to biases in the estimated results. To address the above issues, this study uses the propensity score matching method to find the most similar control group enterprises for each city in the treatment group and then uses the matched samples for model estimation. Propensity scores were calculated for the covariables. Because there are fewer samples in the treatment group and more matching variables, to avoid information loss caused by the failure of too many sample matches, 1:4 nearest neighbor kernel density matching based on logit regression is adopted, and regression analysis is carried out on the matched samples. The matching results are presented in Table 9 and Fig 6. After matching, the variables were concentrated around zero, and the biases were all approximately 10%. The P values were significant before matching and were not significant after matching. The results show

**Table 9. The results of the balance test.**

| Variable | Unmatched | Mean | | %bias | t-test | |
|---|---|---|---|---|---|---|
| | Matched | Treated | Control | | t | p > \|t\| |
| GNP | U | 14.268 | 5.786 | 273.100 | 12.070 | 0.000 |
| | M | 13.485 | 14.287 | −10.100 | −0.730 | 0.468 |
| Pdensity | U | 7.478 | 5.730 | 187.100 | 8.530 | 0.000 |
| | M | 7.128 | 7.253 | −13.400 | −0.380 | 0.705 |
| Dopen | U | 0.625 | 0.175 | 152.400 | 7.820 | 0.000 |
| | M | 0.530 | 0.515 | 5.300 | 0.140 | 0.891 |
| Vpostal | U | 480.400 | 18.215 | 137.100 | 35.880 | 0.000 |
| | M | 207.790 | 274.210 | −1.000 | −0.040 | 0.968 |
| Vtele | U | 349.150 | 47.038 | 186.600 | 18.400 | 0.000 |
| | M | 288.430 | 267.180 | 13.100 | 0.290 | 0.777 |
| Npetent | U | 15.162 | 1.007 | 180.300 | 26.310 | 0.000 |
| | M | 11.935 | 10.517 | 18.100 | 0.440 | 0.663 |
| DFiscal | U | 0.780 | 0.448 | 206.500 | 7.780 | 0.000 |
| | M | 0.794 | 0.780 | −3.600 | −0.130 | 0.894 |

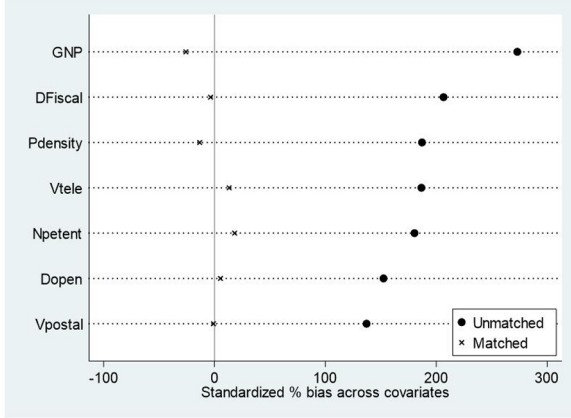

**Fig 6. Graph of the standardized deviation of the variables.**

that, after the above matching, the treatment and control groups show significant approximation and coverage. Therefore, the control group samples obtained through propensity score matching were considered valid. Finally, the DID estimation was carried out again using the matched samples. The regression results are shown in Column (1) of Table 10. Consistent with the previous baseline regression results, the regression coefficient of AI application is still significantly positive; that is, the urban innovation performance of the policy pilot cities has significantly improved, indicating that the results of the above baseline regression are reliable.

**Placebo tests.**

Temporal Placebo Test: A temporal placebo test was employed to ensure that the differences in urban innovation performance between pilot and non-pilot cities were not caused by temporal changes. In this study, the policy of the pilot zone for innovative applications of AI was advanced by four, five, and six years in terms of time, and false policy time points were constructed. Regressions were then conducted for these false-policy time points. The results, as shown in Columns (2)–(4) of Table 10, indicate that on the premise of controlling other influencing factors, the estimated regression coefficients of AI_Policy all failed the test, regardless of the specific false policy time used. This suggests that there is no systematic difference in the temporal trends of the pilot zone for innovative applications of AI and, once again, affirms that the application of AI effectively promotes urban.

Urban Placebo Test: To avoid the impact of unobservable omitted variables on the regression results, an urban placebo test was conducted by replacing the cities in the treatment group. In this study, some cities were randomly selected from the sample cities as false treatment group cities, and the remaining cities were regarded as control group cities. Thus, we can obtain the estimated coefficients of the impact of AI applications on urban innovation performance under the implementation of an urban placebo. The above process was repeated 500 times, and the kernel density and distribution of the P values of the 500 estimated coefficients are presented in Fig 7. The results show that the estimated coefficients generated during the random treatment process are mainly concentrated at approximately 0. Thus, the relationship between AI applications and urban innovation performance is not caused by unobservable factors, and the conclusions of this study have a certain degree of robustness.

**IV treatment.** In the baseline regression, some control variables were selected, and both city fixed effects and time fixed effects were controlled to reduce the impact of other factors on urban innovation performance. However, omitted variables may still exist, leading to biases

**Table 10. Robustness test.**

| Methods | PSM-DID | Time placebo test | | | IV | |
|---|---|---|---|---|---|---|
| Variable | UIP | Pre4 | Pre5 | Pre6 | AI_Policy | UIP |
| Columns | (1) | (2) | (3) | (4) | (5) | (6) |
| AI_Policy | 1.579*** | 0.827 | 0.754 | 0.855 | | 4.267*** |
| | (0.388) | (0.512) | (0.622) | (0.596) | | (0.931) |
| PT×AI-Enterprise | | | | 0.031*** | | |
| | | | | (0.002) | | |
| Control | YES | YES | YES | YES | YES | YES |
| City FE | YES | YES | YES | YES | YES | YES |
| Year FE | YES | YES | YES | YES | YES | YES |
| Observations | 173 | 3,026 | 3,026 | 3,026 | 3,026 | 3,026 |

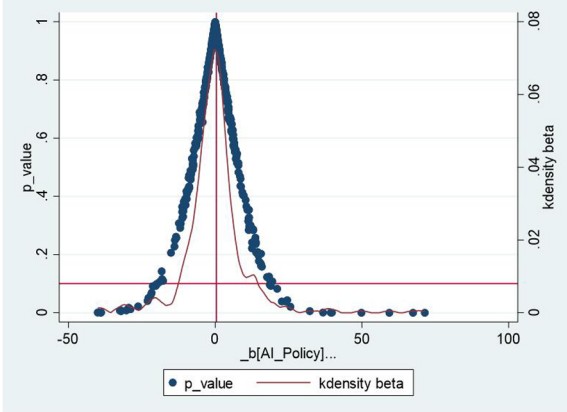

**Fig 7. Placebo test diagram.**

in the regression results. Therefore, to address the potential bias caused by other unobservable factors, this study selects the interaction term between the number of local telephone subscribers at the end of the year in each prefecture-level city (PT) and the stock of artificial intelligence enterprises (AI-Enterprises) as an instrumental variable to mitigate possible endogeneity problems. After testing, the instrumental variable satisfied the exogeneity condition. Columns (5)–(6) of Table 10 show the regression results between the IV variable and the core explanatory variable. The estimated coefficient of the IV variable is significantly positive at the 1% level and passes the weak instrumental variable test. Since the number of instrumental variables is equal to the number of endogenous variables, both being one, an over-identification test cannot be conducted. This indicates a positive correlation between the IV variable and AI_Policy, which is consistent with expectations. Moreover, in Column (5), the estimated coefficient of AI_Policy is still significantly positive, suggesting that, after the instrumental variable is used to control for the endogeneity problem, the original results still hold.

**Other robustness tests.**

Eliminating Interference from Other Policies: During the investigation period of this study, the smart city pilot policies established in three batches after 2012 and the special action of the intelligent manufacturing pilot demonstration proposed in 2015 are closely related to the research presented in this paper. Therefore, in the baseline regression model, we incrementally add dummy variables for both the Smart City Pilot Policy and the Smart Manufacturing Demonstration Project Special Action to mitigate their impact on the estimation results. The findings are presented in Columns (1) and (2) of Table 11. The estimation results show that, after adding the two control policies, the application of AI still has a significant effect on urban innovation performance, and the results remain robust.

Change in the Sample Interval: This study retains only the data after the year before the policy was implemented; that is, the data from 2018–2023 are used to conduct the regression again. According to the results shown in Column (3) of Table 11, after the time span is shortened, the coefficient of AI_Policy remains significant, which, to a certain extent, indicates that the basic regression results of this study are robust.

**Table 11. Other robustness tests.**

| Variable | UIP | | | |
|---|---|---|---|---|
| | *Interference from Other Policies* | | *Sample Interval Change* | *Winsorization* |
| Methods | *Smart City* | *Intelligent Manufacturing* | *2018-2023* | *Cut(1,99)* |
| Columns | (1) | (2) | (3) | (4) |
| AI_Policy | 2.188*** | 1.973*** | 1.259*** | 2.568*** |
| | (0.084) | (0.085) | (0.061) | (0.083) |
| Smart City | −0.043 | | | |
| | (0.040) | | | |
| Intelligent Manufacturing | | 0.801*** | | |
| | | (0.074) | | |
| Control | YES | YES | YES | YES |
| City FE | YES | YES | YES | YES |
| Year FE | YES | YES | YES | YES |
| Observations | 3,026 | 3,026 | 1,378 | 3,026 |
| R-squared | 0.598 | 0.614 | 0.507 | 0.552 |

Winsorization Treatment: To eliminate possible outliers in the variables, this study conducts a two-sided winsorization treatment on all the control variables, and the results are shown in Column (4) of Table 11. As Table 11 shows, after the winsorization of the variables, the regression coefficient of AI_Policy is still significantly positive at the 1% level, and the original regression results still hold.

**Mechanism testing.**

Testing the Effect of Human Capital Upgrading. According to previous analyses, the application of AI can promote the upgrading of human capital and improve urban innovation performance. First, considering the impact of the establishment of AI applications on human capital upgrading, the regression results in Column (1) of Table 12 indicate that the application of AI has a promoting effect on human capital upgrading. Second, considering the promoting effect of human capital upgrading as a mediating variable, Column (2) of Table 12 shows that the coefficient of AI application is significantly positive at the 1% level and that the coefficient is smaller than the baseline regression coefficient (1.983 < 2.189), indicating that the application of AI can promote urban innovation performance through human capital upgrading and that human capital upgrading has a mediating effect, which verifies Hypothesis 2.

Further analysis of the impact of human capital upgrades on the "quality" and "quantity" of urban innovation performance, as shown in Columns (3) and (4) of Table 12, reveals that the coefficient of AI application is significantly positive at the 1% level and smaller than the baseline regression coefficient (1.131 < 1.663; 1.024 < 1.189), and the coefficient of human capital upgrades is significantly positive, indicating that human capital upgrades play a positive role in the positive impact of AI application on both the quality and quantity of urban innovation and have a stronger impact on the quantity of urban innovation.

Test of the Effect of Absorptive Capacity Enhancement. The application of AI can enhance the knowledge absorptive capacity of a city, thereby improving urban innovation performance. First, we consider the impact of AI applications on the absorption capacity of a city. As shown in Column (1) of Table 13, the application of AI can significantly promote absorptive capacity. We then consider the mediating effect of absorptive capacity. As shown in Column (2) of Table 13, the coefficient of AI application is significantly positive at the 1% significance level, and the coefficient is smaller than the coefficient of the baseline regression (1.997<2.189), indicating that the application of AI can further promote the improvement of

**Table 12. The effect of human capital upgrading.**

| Variable | Lhuca | UIP | Iquantity | Iquality |
|---|---|---|---|---|
| Columns | (1) | (2) | (3) | (4) |
| AI_Policy | 0.623*** | 1.983*** | 1.131*** | 1.024*** |
| | (0.094) | (0.091) | (0.114) | (0.115) |
| Lhuca | | 0.490*** | 0.345*** | 0.314*** |
| | | (0.021) | (0.026) | (0.026) |
| Control | YES | YES | YES | YES |
| City FE | YES | YES | YES | YES |
| Year FE | YES | YES | YES | YES |
| Constant | 0.090*** | −0.013 | −0.025 | −0.280*** |
| | (0.025) | (0.024) | (0.030) | (0.037) |
| Observations | 2,504 | 2,504 | 2,504 | 2,503 |
| R-squared | 0.329 | 0.618 | 0.283 | 0.672 |
| Number of id | 274 | 274 | 274 | 273 |

**Table 13. The effect of absorptive capacity enhancement.**

| Variable | Cabsorp | UIP | Iquantity | Iquality |
|---|---|---|---|---|
| Columns | (1) | (2) | (3) | (4) |
| AI_Policy | 0.427*** | 1.997*** | 1.594*** | 1.094*** |
| | (0.075) | (0.078) | (0.114) | (0.110) |
| Cabsorp | | 0.449*** | 0.161*** | 0.224*** |
| | | (0.020) | (0.029) | (0.029) |
| Control | YES | YES | YES | YES |
| City FE | YES | YES | YES | YES |
| Year FE | YES | YES | YES | YES |
| Constant | 0.038* | −0.009 | 0.044 | −0.237*** |
| | (0.020) | (0.020) | (0.030) | (0.029) |
| Observations | 3,026 | 3,026 | 3,026 | 3,016 |
| R-squared | 0.528 | 0.662 | 0.314 | 0.716 |
| Number of id | 280 | 280 | 280 | 280 |

urban innovation performance by enhancing absorptive capacity. Absorptive capacity has a mediating effect, and Hypothesis 3 holds.

We further analyze the impact of absorptive capacity on the "quality" and "quantity" of urban innovation performance. The regression results are shown in Columns (3) and (4) of Table 13. The coefficient of AI application is significantly positive at the 1% significance level and is smaller than the coefficient of the baseline regression (1.594 < 1.663; 1.094 < 1.189), and absorptive capacity is significantly positive. This shows that absorptive capacity has a mediating effect on the impact of AI application in promoting the quality and quantity of urban innovation.

Moderating Effect of the Degree of Government Intervention. As shown in Table 14, the coefficient of the interaction term between the degree of government intervention and the application of AI has a sign opposite to that of the coefficient of AI application. This finding indicates that the degree of government intervention has a significant negative moderating effect on the positive effect of AI applications on urban innovation performance. As the degree of government intervention increases, the weakening effect of AI on urban innovation performance becomes more significant. A further examination of the impact of government intervention on the quality and quantity of urban innovation, as shown in columns (2) and (3) of Table 14, reveals that there is a significant weakening effect in both cases, thus verifying Hypothesis 4.

**Table 14. The effect of the degree of government intervention.**

| Variable | UIP | Iquantity | Iquality |
|---|---|---|---|
| Columns | (1) | (2) | (3) |
| Dinter×AI_Policy | 0.297*** | −0.140*** | −0.342*** |
| | (0.029) | (0.039) | (0.038) |
| AI_Policy | −0.901*** | 3.117*** | 4.748*** |
| | (0.307) | (0.423) | (0.407) |
| Dinter | 0.330* | 0.504* | 0.546** |
| | (0.198) | (0.273) | (0.263) |
| Control | YES | YES | YES |
| City FE | YES | YES | YES |
| Year FE | YES | YES | YES |
| Constant | −0.032 | −0.040 | −0.338*** |
| | (0.039) | (0.054) | (0.052) |
| Observations | 3,026 | 3,026 | 3,016 |
| R-squared | 0.614 | 0.310 | 0.718 |

**Heterogeneity analysis.**

Locational Heterogeneity. The new economic geography proposed by Krugman et al. points out that the foundation and scale of a regional economy affect the agglomeration of various production factors. Regions with a better economy often have a larger scale of industrial and factor agglomeration, resulting in agglomeration and scale effects. The essence of regional innovation is the process of moving from innovation input to output. Influenced by various factors, such as economic foundation, the promoting effect of AI applications on urban innovation performance varies across different regions. In this study, the research samples were divided from the perspective of the Yangtze River Economic Belt to further compare the characteristics of regional differences in the impact of AI applications on urban innovation performance. As shown in Columns (1) and (2) of Table 15, AI applications are significant only in the cities of the Yangtze River Economic Belt. This is because, as one of the relatively developed economic regions in China, the Yangtze River Economic Belt has a strong economic foundation and scale and has significant advantages in elements such as capital, technology, and talent, providing strong support for the promotion of policies for innovative applications of AI. Moreover, the urban agglomeration in the Yangtze River Economic Belt has a significant industrial agglomeration effect, forming a complete industrial cluster in the technological field, which promotes technological spillover and knowledge sharing. This advantage accelerates the transformation of innovation. In contrast, cities outside the Yangtze River Economic Belt have a relatively weak economic foundation and insufficient resource endowment, making it difficult to form an effective innovation ecosystem and resulting in insignificant effects of AI applications.

Handling of Temporal Heterogeneity. The policies of the pilot zone for the innovative application of AI were implemented in three batches in 2019, 2020, and 2021. To refine this study, the urban performance related to the policies of the pilot zones for the innovative application of AI was decomposed into the above three groups, and the results are shown in Columns (3)–(5) of Table 15. According to the coefficient of AI_Policy, the policies of the pilot zones for the innovative application of AI in all three batches had a positive effect on urban innovation performance, among which the policies implemented in 2019 had the strongest effect. This may be due to the selection of policy pilots. Typically, the cities that are most capable of implementing AI policies are included in the first batch of Pilot Zones for the Innovative Application of AI. Additionally, although the cities in the control group were

**Table 15. Heterogeneity analysis.**

| Variable | UIP | | | | | | |
|---|---|---|---|---|---|---|---|
| Groups | Region | | Time | | | Urban scale | |
| | Yangtze River | Non-Yangtze | 2019 | 2020 | 2021 | Super-large Cities | Megacities |
| Columns | (1) | (2) | (3) | (4) | (5) | (6) | (7) |
| AI_Policy | 1.727*** | −0.039 | 1.268*** | 0.528** | 0.741 | 0.932 | 0.556*** |
| | (0.076) | (0.100) | (0.133) | (0.229) | (0.572) | (0.879) | (0.133) |
| City FE | YES | YES | YES | YES | YES | YES | YES |
| Year FE | YES | YES | YES | YES | YES | YES | YES |
| Control | YES | YES | YES | YES | YES | YES | YES |
| Observations | 1,152 | 1,874 | 3,026 | 3,026 | 3,026 | 77 | 154 |

not included in the pilot projects, they imitated the first batch of pilot cities in implementing AI applications, which led to a decrease in the impact of the policies in the second and third batches.

Heterogeneity of City Sizes. The application of AI not only broadens the channels for innovation entities to access cutting-edge technologies but also expands the scope of promotion of technology providers. City size is a crucial factor that influences the effectiveness of policies. On the one hand, it determines the scope and quality of resources that enterprises in a city can obtain, thereby affecting the innovation performance of the city. However, it also impacts the allocation and efficiency of urban innovation resources, which in turn affects urban innovation performance. Therefore, it is necessary to further explore the impact of AI applications on urban innovation performance for cities of different sizes. In existing research, city sizes are classified into four categories on the basis of the year-end population: Super-large Cities > Megacities > Large Cities > Medium-sized and Small Cities. Since the policy of the pilot zone for the innovative application of AI is piloted only in super-large cities and megacities, only these two types of cities are compared in terms of policy effects. According to the results in Columns (6) and (7) of Table 15, there is significant heterogeneity in city size regarding the impact of the policy on urban innovation performance. Specifically, implementing this policy in megacities has a significant positive effect on the quality of urban innovation, whereas for super-large cities, there is a reverse inhibitory effect, although it is not significant. This is because innovation entities are concentrated in super-large cities and the demand for technology is strong, which can accelerate the implementation of policies. However, resource allocation is prone to imbalances. Moreover, the industrial structure of super-large cities tends to be mature and rigid, resulting in policy effects that do not meet expectations.

## Conclusions and implications

Enhancing innovation is an inherent requirement for withstanding risks and improving competitiveness. The Chinese government introduced several policy measures to this end. On the basis of a quasi-natural experiment involving pilot zones for the innovative application of AI and on the basis of theoretical analysis, this study uses panel data from 283 cities from 2012 to 2023 and adopts methods such as multi-period DID to empirically test the impact of the application of AI on urban innovation performance. The main conclusions are as follows: AI significantly improves urban innovation performance and simultaneously promotes both the "quality" and "quantity" of urban innovation. The upgrading of human capital and the improvement of absorptive capacity are important channels through which the application of AI affects urban innovation performance, and the degree of government intervention plays

a restraining role in the influence of these two factors. In the Yangtze River Economic Belt, megacities and regions where policies were implemented in 2019, the effect of the application of AI in promoting urban innovation performance is more obvious. The following policy implications can be drawn from the conclusions of this study:

First, the development and application of AI technologies should be strengthened. While supporting breakthrough research in the basic theories of AI, inter-disciplinary and cutting-edge research projects should be encouraged to fundamentally promote the original innovation of AI technologies and lay a foundation for their applications. Furthermore, we would build a collaborative innovation ecosystem, establish an "AI Industry-University-Research Alliance," and AI industrial parks are created to reduce the R&D costs of enterprises, affect industrial clusters, and stimulate innovation vitality.

Second, strengthen the talent training system and optimize the recruitment mechanism. High-quality personnel and good absorption capabilities are crucial for promoting the development of AI. We should enhance the cultivation and recruitment of talent in artificial intelligence, digital fields, and related areas. We must improve the talent development system, both through internal training programs and external recruitment initiatives. This will boost the workforce's AI technical capabilities and knowledge absorption capacity, providing essential support for the advancement of AI technology and urban innovation.

Third, the government should take initiative to participate and provide active and reasonable guidance. The intervention role of the "visible hand" should be reasonably considered, and the inappropriate intervention of local government finances in innovative industries should be reduced. The central government should introduce policies to guide local governments in formulating reasonable fiscal and industrial policies to strongly support the development of innovative industries. It is necessary to avoid local government intervention by distorting the market and enhance the status of the market in the allocation of resources and factors. Simultaneously, local governments need to introduce relevant measures and policies to transform the "race to the bottom" approach in the process of formulating fiscal policies into an "innovation-driven" approach.

Fourth, pilot zones for the innovative application of AI should be constructed in accordance with local conditions. The flexibility and inclusiveness of policy implementation should be strengthened. In cities with a high level of science, technology, and education, it is necessary to continue optimizing the urban innovation environment, actively guide the transformation of scientific and technological achievements of pilot smart cities, and leverage the policies of pilot zones to build a characteristic urban innovation ecosystem, thereby promoting the high-quality development of cities.

## Supporting information

**S1 Appendix. Specific situations of the construction of the pilot zones for the innovative application of AI.**
(PDF)

**S2 File. Minimal data set.**
(XLSX)

## Author contributions

**Conceptualization:** Shuang Han.

**Data curation:** Shuang Han.

**Formal analysis:** Shuang Han.

**Funding acquisition:** Xianmin Sun.

**Investigation:** Shuang Han.

**Methodology:** Shuang Han, Shen Zhong.

**Project administration:** Xianmin Sun, Shen Zhong.

**Supervision:** Shen Zhong.

**Validation:** Shuang Han, Shen Zhong.

**Visualization:** Shuang Han.

**Writing – original draft:** Shuang Han.

**Writing – review & editing:** Shuang Han.

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
