## [Decision Letter · Decision Letter 0]

21 May 2025

PONE-D-25-20547Applications of Artificial Intelligence and Urban Innovation Performance: A Quasi-natural Experiment Based on the Pilot Zones for the Innovative Application of Artificial IntelligencePLOS ONE

Dear Dr. Sun,

Thank you for submitting your manuscript to PLOS ONE. After careful consideration, we feel that it has merit but does not fully meet PLOS ONE’s publication criteria as it currently stands. Therefore, we invite you to submit a revised version of the manuscript that addresses the points raised during the review process.

We look forward to receiving your revised manuscript.

Kind regards,

Taiyi He

Academic Editor

PLOS ONE

Journal Requirements:

3. Please upload a copy of Supporting Information Figure/Table/etc. “Supporting information” which you refer to in your text on page 23 and 24.

Reviewers' comments:

Reviewer's Responses to Questions

**Comments to the Author**

1. Is the manuscript technically sound, and do the data support the conclusions?

Reviewer #1: Yes

Reviewer #2: Yes

2. Has the statistical analysis been performed appropriately and rigorously? 

Reviewer #1: Yes

Reviewer #2: Yes

3. Have the authors made all data underlying the findings in their manuscript fully available?

Reviewer #1: Yes

Reviewer #2: Yes

4. Is the manuscript presented in an intelligible fashion and written in standard English?

Reviewer #1: Yes

Reviewer #2: Yes

5. Review Comments to the Author

Reviewer #1: This article takes the panel data of 283 cities in China as samples and systematically studies the impact of artificial intelligence (AI) application on urban innovation performance and its mechanism of action. The topic has certain practical significance. Generally speaking, the content of this article is detailed, the method is scientific, and the conclusion is reliable, meeting the publication standards of the journal. I suggest accepting it.

Reviewer #2: 1. in line 330� “Currently, there are mainly four types of indicators for measuring urban innovation

performance" ,but the author only wrote three in the narrative, do you need to change it?

2. There is a problem with the city-fixed and time-fixed corner markers at the back of the formulae

3. Parallel Trend Chart Pressure Lines for Benchmark Regression to be Adjusted

6. PLOS authors have the option to publish the peer review history of their article (what does this mean?). If published, this will include your full peer review and any attached files.

Reviewer #1: No

Reviewer #2: No

---

## [Author Response · Author response to Decision Letter 1]

6 Jun 2025

Dear Dr. Taiyi He and the PLOS ONE Editorial Team,

Thank you very much for the detailed and constructive feedback on our manuscript titled “Applications of Artificial Intelligence and Urban Innovation Performance: A Quasi-natural Experiment Based on the Pilot Zones for the Innovative Application of Artificial Intelligence” (Manuscript ID: PONE-D-25-20547). We truly appreciate the time and effort put into reviewing our work.

We have carefully considered the comments and suggestions from the reviewers and have made the necessary revisions to improve the quality and clarity of our manuscript. Below is a summary of the changes we have made in response to the reviewers’ comments:

1.Revisions to the Text:

1)We have supplemented the fourth method for measuring urban innovation performance and conducted a comparative analysis.

2)We have corrected the issue with the city-fixed and time-fixed corner markers in the formulae.

3)We have adjusted the Parallel Trend Chart Pressure Lines for Benchmark Regression to ensure accuracy and clarity.

2.Supporting Information:

We have uploaded the copy of Supporting Information Figure/Table/etc. “Supporting information” as requested.

3.Reference List:

We have reviewed and ensured that our reference list is complete and correct. We have removed any retracted references and replaced them with relevant current references.

4.ORCID iD:

We have ensured that the corresponding author has an ORCID iD and that it is validated in Editorial Manager.

5.Manuscript Formatting:

We have reviewed and ensured that our manuscript meets PLOS ONE’s style requirements, including those for file naming.

6.Rebuttal Letter and Marked-up Manuscript:

1)We have prepared a detailed rebuttal letter that responds to each point raised by the academic editor and reviewer(s). This letter is uploaded as a separate file labeled “Response to Reviewers.”

2)We have also prepared a marked-up copy of the manuscript that highlights changes made to the original version. This file is labeled “Revised Manuscript with Track Changes.”

3)Additionally, we have uploaded an unmarked version of the revised manuscript labeled “Manuscript.”

We are confident that these revisions have significantly improved the manuscript and have addressed the reviewers’ concerns. We look forward to your positive response and the opportunity to contribute to PLOS ONE.

Thank you once again for your patience and understanding. If you have any further questions or require additional information, please do not hesitate to contact us.I am always looking forward to receiving any suggestions from you. Your suggestions will make our research more perfect. Thank you very much.Wishing you smooth progress in your work.

Best regards!

---

## [Decision Letter · Decision Letter 1]

23 Jun 2025

PONE-D-25-20547R1Applications of Artificial Intelligence and Urban Innovation Performance: A Quasi-natural Experiment Based on the Pilot Zones for the Innovative Application of Artificial IntelligencePLOS ONE

Dear Dr. Sun,

Thank you for submitting your manuscript to PLOS ONE. After careful consideration, we feel that it has merit but does not fully meet PLOS ONE’s publication criteria as it currently stands. Therefore, we invite you to submit a revised version of the manuscript that addresses the points raised during the review process.

We look forward to receiving your revised manuscript.

Kind regards,

Taiyi He

Academic Editor

PLOS ONE

Journal Requirements:

Additional Editor Comments:

The authors are required to check the details carefully and revise the manuscript according to reviewers' comment.

Reviewers' comments:

Reviewer's Responses to Questions

**Comments to the Author**

1. If the authors have adequately addressed your comments raised in a previous round of review and you feel that this manuscript is now acceptable for publication, you may indicate that here to bypass the “Comments to the Author” section, enter your conflict of interest statement in the “Confidential to Editor” section, and submit your "Accept" recommendation.

Reviewer #1: All comments have been addressed

Reviewer #2: All comments have been addressed

2. Is the manuscript technically sound, and do the data support the conclusions?

Reviewer #1: Yes

Reviewer #2: Yes

3. Has the statistical analysis been performed appropriately and rigorously? 

Reviewer #1: Yes

Reviewer #2: Yes

4. Have the authors made all data underlying the findings in their manuscript fully available?

Reviewer #1: Yes

Reviewer #2: Yes

5. Is the manuscript presented in an intelligible fashion and written in standard English?

Reviewer #1: Yes

Reviewer #2: Yes

6. Review Comments to the Author

Reviewer #1: The author has carefully revised the manuscript and believes it is now polished and ready for publication, meeting all necessary standards and requirements.

Reviewer #2: The author has made substantial revisions based on my suggestions, and I recommend acceptance of the manuscript. However, the author needs to incorporate all figures into the main text. Currently, all figures are provided only as supplementary materials and should be embedded within the article. Therefore, I recommend minor revision. Acceptance is recommended upon implementation of this amendment.

7. PLOS authors have the option to publish the peer review history of their article (what does this mean?). If published, this will include your full peer review and any attached files.

Reviewer #1: No

Reviewer #2: No

---

## [Author Response · Author response to Decision Letter 2]

26 Jun 2025

Dear Editor:

I have completed the revision in strict accordance with the requirements. Currently, three decimal places are retained in the tables of the main text, and robust standard errors are additionally marked. All figures have been re-uploaded to the designated positions in the submission system. This revision has effectively standardized the submission format. Thank you for your careful review and the valuable opportunity to revise the manuscript. If there are any other areas that need adjustment, please feel free to inform me, and I will fully cooperate to improve them!

Yours sincerely

---

## [Editor Report · Decision Letter 2]

11 Jul 2025

PONE-D-25-20547R2Applications of Artificial Intelligence and Urban Innovation Performance: A Quasi-natural Experiment Based on the Pilot Zones for the Innovative Application of Artificial IntelligencePLOS ONE

Dear Dr. Sun,

Thank you for submitting your manuscript to PLOS ONE. After careful consideration, we feel that it has merit but does not fully meet PLOS ONE’s publication criteria as it currently stands. Therefore, we invite you to submit a revised version of the manuscript that addresses the points raised during the review process.

We look forward to receiving your revised manuscript.

Kind regards,

Taiyi He

Academic Editor

PLOS ONE

Journal Requirements:

**Additional Editor Comments:**

Please carefully check the formula spelling and instruction, and the author need to polish the language.

---

## [Author Response · Author response to Decision Letter 3]

19 Jul 2025

Dear Editor,

I have completed the revisions in strict accordance with the requirements. Currently, all formulas and formula explanations in the main text of the article have been corrected to ensure accuracy. Furthermore, in accordance with the requirements, this study has been polished, and this revision has effectively standardized the submission format. Thank you very much for your professional guidance and valuable revision opportunities. If there are any other areas that need to be adjusted, please let me know, and I will do my best to improve them.

---

## [Editor Report · Decision Letter 3]

22 Jul 2025

Applications of Artificial Intelligence and Urban Innovation Performance: A Quasi-natural Experiment Based on the Pilot Zones for the Innovative Application of Artificial Intelligence

PONE-D-25-20547R3

Dear Dr. Sun,

We’re pleased to inform you that your manuscript has been judged scientifically suitable for publication and will be formally accepted for publication once it meets all outstanding technical requirements.

Kind regards,

Taiyi He

Academic Editor

PLOS ONE

Additional Editor Comments (optional):

I am satisfied with the revision and this manuscript can be accepted for plos one.
---

## [Editor Report · Acceptance letter]

PONE-D-25-20547R3

PLOS ONE

Dear Dr. Sun,

I'm pleased to inform you that your manuscript has been deemed suitable for publication in PLOS ONE. Congratulations! Your manuscript is now being handed over to our production team.

Kind regards,

on behalf of

Dr. Taiyi He

Academic Editor

PLOS ONE